# Effectiveness of a suction device for containment of pathogenic aerosols and droplets

Kai Lordly[1], Ahmet E. Karataş[1], Steve Lin[2,3], Karthi Umapathy[4], Rohit Mohindra[3,5,6] *

1 Department of Aerospace Engineering, Toronto Metropolitan University, Toronto, Ontario, Canada,
2 Keenan Research Centre, Li Ka Shing Knowledge Institute, St. Michael's Hospital, Toronto, Ontario,
Canada, 3 Division of Emergency Medicine, Faculty of Medicine, University of Toronto, Toronto, Ontario,
Canada, 4 Department of Electrical, Computer and Biomedical Engineering, Toronto Metropolitan University,
Toronto, Ontario, Canada, 5 Department of Biomedical Engineering, Faculty of Engineering and Architectural
Sciences, Toronto Metropolitan University, Toronto, Ontario, Canada, 6 Schwartz Reisman Emergency
Medicine Institute, Toronto, Ontario, Canada

☉ These authors contributed equally to this work.
* rohit.mohindra@nygh.on.ca

## Abstract

### Background

As the global community begins recovering from the COVID-19 pandemic, the challenges
due to its aftermath remain. This health crisis has highlighted challenges associated with air-
borne pathogens and their capacity for rapid transmission. While many solutions have
emerged to tackle this challenge, very few devices exist that are inexpensive, easy to manu-
facture, and versatile enough for various settings.

### Methods

This paper presents a novel suction device designed to counteract the spread of aerosols
and droplets and be cost-effective and adaptable to diverse environments. We also con-
ducted an experimental study to evaluate the device's effectiveness using an artificial cough
generator, a particle counter, and a mannequin in an isolated system. We measured droplet
removal rates with simulated single and repeated cough incidents. Also, measurements
were taken at four distinct areas to compare its effectiveness on direct plume versus indirect
particle removal.

### Results

The device reduced airborne disease transmission risk, as evidenced by its capacity to
decrease the half-life of aerosol volume from 23.6 minutes to 15.6 minutes, effectively cap-
turing aerosol-sized droplets known for their extended airborne persistence. The suction
device lessened the peak total droplet volume from peak counts. At 22 minutes post peak
droplet count, the count had dropped 24% without the suction device and 43% with the suc-
tion device.

org/10.1371/journal.pone.0305842

University of Medical Science, ISLAMIC REPUBLIC
OF IRAN

**Data Availability Statement:** CAD files and data
files are available at 10.5281/zenodo.11264347.

**Funding:** Dr. Ahmat. E. Karatas received an Alliance grant from the Natural Sciences and Engineering Research Council of Canada.

**Competing interests:** The authors have declared that no competing interests exist.

## Conclusions

The experiment's findings confirm the suction device's capability to effectively remove droplets from the environment, making it a vital tool in enhancing indoor air quality. Given the sustained performance of the suction device irrespective of single or multiple cough events, this demonstrates its potential utility in reducing the risk of airborne disease transmission. 3D printing for fabrication opens the possibility of a rapid iterative design process, flexibility for different configurations, and rapid global deployment for future pandemics.

## Introduction

The COVID-19 pandemic had over 4.9 million known infections in Canada and continues to have lasting impacts on our healthcare system [1]. Throughout the pandemic, healthcare providers were also at high risk of being exposed to COVID-19. In the initial wave alone, healthcare providers accounted for 19.4% of infections [2]. Given that COVID-19 and other respiratory illnesses will continue to have an impact on healthcare providers, it is important to look for innovative devices to control the rapid spread of pathogenic aerosols in closed spaces to avoid such aerosol-triggered pandemics in the future, especially in settings that do not have access to adequate ventilation systems. Numerical simulations have demonstrated that, in a clinical environment, suction devices significantly lower healthcare workers' exposure [3].

Few studies on aerosol removal devices have been recently introduced in the scientific literature [3, 4]. However, there remains a lack of comprehensive data on the performance of these suction devices in terms of spatial and temporal measurements. Furthermore, the literature highlights a significant oversight: the lack of a potentially community-driven, iterative design process that facilitates rapid prototyping and global deployment with minimal resources.

This study introduces several novelties to address these gaps:

1. The development and testing of a novel, rapidly manufacturable suction device designed for iterative improvements. This device is versatile enough to be adapted for use in multiple scenarios.

2. A comprehensive and controlled experimental framework to directly measure droplet concentrations, enabling a more accurate evaluation of suction device efficacy. This approach significantly advances the methodological tools available for such assessments.

In our experimental investigation, we first report on the design of an adaptable, rapidly manufacturable suction canister device. We then conducted comprehensive measurements of droplet concentration across different spatial configurations and quantified the half-lives of droplets within various size categories to demonstrate the device's efficacy.

## Methodology

### Canister design and fabrication

The study introduces a new suction device specifically engineered to mitigate aerosol and droplet generation through high-powered suction. The primary goal is to lower the potential risk of infection in the surrounding environment by minimizing aerosol particle count. After multiple design consultations with two clinicians and fluid dynamics specialists, a scalable canister design was selected to achieve this objective.

The canister is cylindrical with a height of 250 mm and a diameter of 110 mm, featuring chamfered corners with a radius of 40 mm. The canister's total surface area is approximately 0.1 m$^2$. The inside of the canister has a cavity for suction with a skin thickness of 5 mm. At the bottom of the canister, there's an attachment neck standing 50 mm high with a diameter of 50 mm. This neck features a 1-1/2 inch National Pipe Tapered (NPT) threading at the tip to facilitate attachment to flat surfaces such as tables. Additionally, the neck contains a concentric inner hole with a ½ inch NPT threading that can connect to fitting adapters for suction.

The canister design features four identical rectangular trenches on the side surface of the main body, arranged in a circular pattern to ensure 360˚ coverage. The purpose of these trenches is to potentially house the planar UVC lights, a significant component for fomite disinfection. The canister has 368 holes, each with a diameter of 3 mm, thereby providing a total suction area of 2601 mm$^2$. These holes were located on the canister's side and top walls.

Canister models were fabricated using Fused Deposition Modelling (FDM) 3D printing, leveraging its speed and versatility for rapid prototyping and complex shape production. A cartesian FDM printer (ANYCUBIC Chiron model, Shenzhen, China) with a substantial build volume of 400 x 400 x 450 mm was utilized. The models were fabricated from polylactic acid (PLA) thermoplastic polyester through a 0.4 mm nozzle. The printing settings included a layer height of 0.2 mm, line width of 0.4 mm, wall thickness corresponding to 4-line counts, and 40% infill density. The use of 3D printing for fabrication opens up the possibility of a rapid iterative design process, flexibility for different configurations, and fast global deployment for future pandemics. A CAD model and an image of the 3D printed prototype of the canister are presented in Fig 1.

## Droplet generation

To evaluate the suction device's effectiveness in capturing aerosols from respiratory activities, we used an artificial cough generator to create a droplet-laden flow that mimics a cough. This generator produced a flow laden with droplets that simulates a natural cough. The SARS-CoV-2 virus is approximately 0.1–0.5 μm in size. On the other hand, saliva ejecta can start as small as 1 μm and can extend up to magnitudes 100 times larger, with an average size of about 10 μm [5]. Generally, cough dynamics are bifurcated into two main categories: large, inertia-dominated droplets (≥10 μm) and puff cloud aerosols (≤10 μm). Droplets sizing 100 μm or more are too heavy for their size to stay airborne, whereas aerosols ranging between 5 and 10 μm remain suspended in the air, as delineated by the Wells Curve [5, 6]. Large droplets tend to follow a ballistic trajectory, whereas aerosol sized droplets circulate within a bulk puff cloud [7] However, it is crucial to understand that the transition from aerosol to droplet is not abrupt, and size is not the sole determinant of cough dynamics. Relying exclusively on the Wells Curve for infection control can lead to misjudgements, especially since the 2-metre distancing recommendation overlooks the turbulent eddy cloud transporting the droplets and instead zeroes in on droplet size alone [8]. In this study, for clarity, both aerosols and droplets will be collectively referred to as "droplets" since we are measuring particle numbers.

The details of this experimental setup have been previously outlined in [9]; thus, only a brief description is provided here. The artificial cough generator was connected to pressurized air lines, the pressure of which was regulated. The maximum cough flow rate was managed by an acrylic flow controller fitted with a needle valve. Aspects of the cough release, such as duration, were controlled by solenoid valves, the opening timings of which were managed by a digital delay generator (Model 575, Berkley Nucleonics, USA). A droplet generator (LaVision, Bielefeld, Germany) was attached to the solenoid valves' outlet to seed the flow with a polydisperse distribution of droplets. A propylene glycol solution was atomized into the flow to

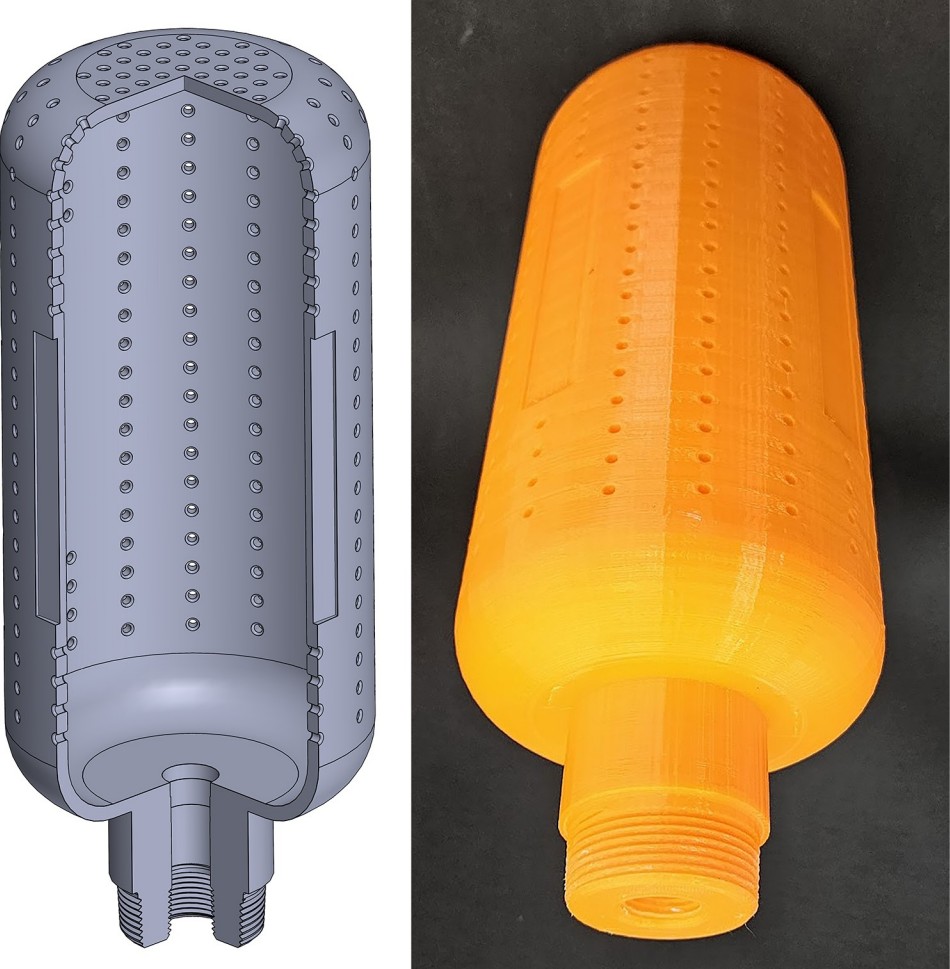

**Fig 1. A sectional view of the CAD model (left) and an image of the 3D printed prototype (right) of the canister model.**

simulate airway secretions. A bypass airflow and an integrated pressure regulator allowed partial control of the droplet loading and size distribution. Finally, the flow outlet was connected to the thoracic cavity of an airway manikin (Ht-Man, Hawktree, Ottawa, Canada), from which the artificial cough was released. We used a commercially available particle counter (Kanomax, New Jersey, USA) to collect samples across six size bins: 0.3, 0.5, 1.0, 3.0, 5.0, and 10.0+ μm. The particle counter was equipped with an integrated pump and operated at a sampling rate of 0.14 Hz.

Unlike the approach in [9], the particle counter in our study was used to measure free space droplet counts and evaluate the suction device's efficacy by directly sampling from within the suction cavity. Given that there are only a few studies on the efficacy of suction devices, the experimental methodology in this field is not yet well-established. For instance, a study [4] employed particle image velocimetry to analyze general flow structures and used water-sensitive filter paper to detect the presence and settling of droplets. In contrast, our study emphasizes direct sampling from specific locations to assess the temporal variation of suspended droplets.

### Droplet measurement

Several experiments were performed to assess the droplet containment efficacy of the suction device. The experimental equipment was set up inside a sealed enclosure, which was 2.5 m long, 1.6 m wide, and 1.9 m tall. The orientation of the airway manikin was adjusted using an articulated arm, where the mouth was fixed at 1 m height from the ground.

The first two experiments evaluated the suction device's capacity to capture airborne particles from the surrounding environment as designed. In the first configuration, the suction device was operated at a flow rate of 400 L/min. Simultaneously, an artificial cough generator created a single cough lasting 500 ms, positioned 50 cm away from the suction device, aligned with the manikin's face. The particle counter, equipped with a 6.35 mm diameter and 30 mm long sampling probe, was used for measurements. A small hole was drilled just above the neck of the canister (50 mm from the base), and the sampling probe was inserted into this hole to measure droplet loading within the suction stream. The particle counter, sampling the flow at 6-second intervals and a suction rate of 2.83 L/min, outputted particle counts as $\#/m^3$. By knowing the suction flow rate generated by the canister and the individual sampling duration, it was possible to calculate the particle removal efficacy of the device. In the second configuration, we maintained the same conditions as before, with one key modification: instead of a single cough event, an artificial cough was repeated every minute following the initial cough at the 5-minute mark within the measurement domain.

In the third and final series of experiments, the particle counter was positioned at angles of 0, 45, 90, and 135 degrees relative to the manikin, as depicted in Fig 2. Unlike previous configurations, the particle counter probe was no longer inserted into the suction canister. Instead, it

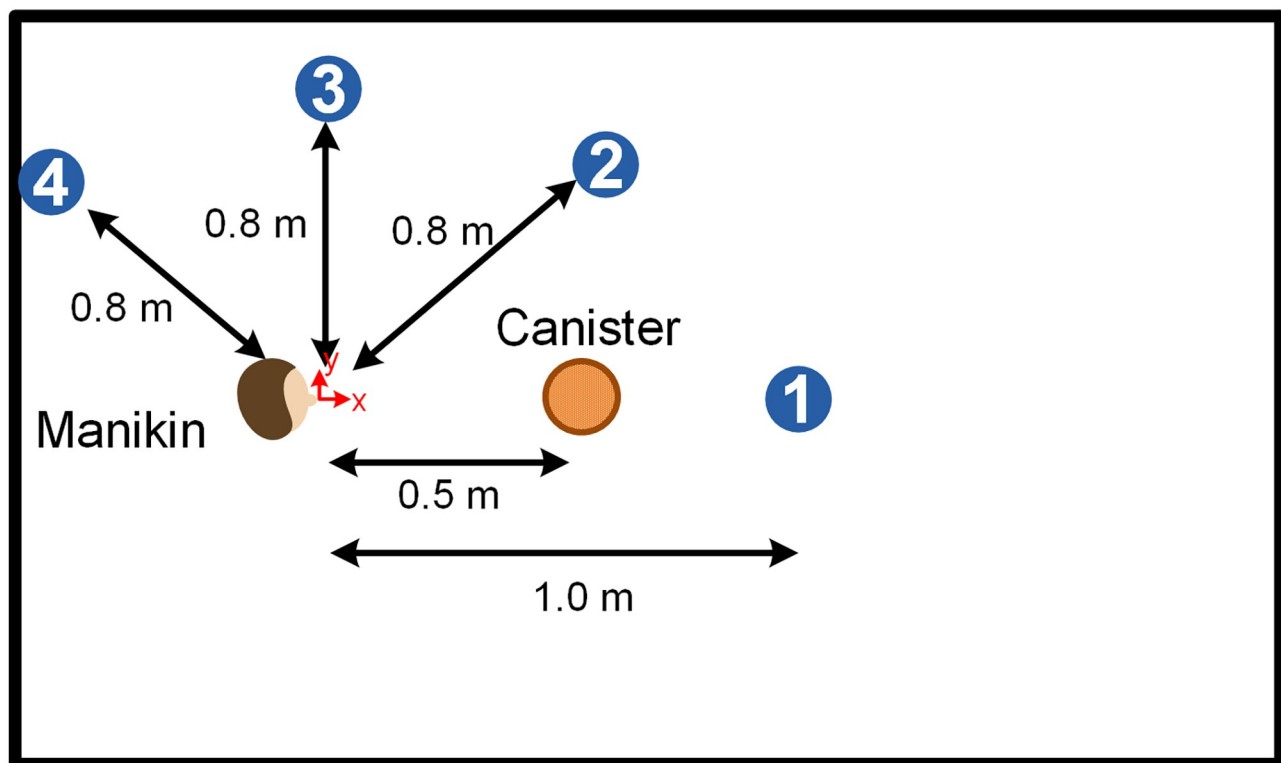

**Fig 2. A schematic of the cough chamber and the measurement locations in relation to the manikin.** The suction device location is fixed at 50 cm away from the manikin.

directly sampled the atmosphere within the cough chamber. This alteration provided a broader perspective on the distribution and behaviour of the droplets post-cough. We kept the suction canister's location constant throughout these experiments and placed it 50 cm from the manikin. This controlled positioning allowed us to focus on the influence of varying the particle counter's location, contributing to a more comprehensive understanding of the suction device's overall droplet containment capabilities.

These experimental approaches collectively provided direct counts of particles captured directly by the suction device and those in the spatial and temporal environmental aerosol load under controlled conditions during single and repeated cough events.

## Outcomes

The primary outcome for the device efficiency is the droplet removal rate, determined by sampling air from inside the suction canister. The droplet removal rate quantifies the number of droplets the canister eliminates per second. This is based on readings from the particle counter, estimated from the duration of particle counter sampling, the suction flow rate of the particle counter, the total suction flow rate of the canister device, and droplet counts from the particle counter itself.

## Results

The droplet removal rate from a single cough event over 35 minutes for droplet size bins of 0.3, 0.5, and 1.0 μm is illustrated in Fig 3. The droplet removal rate, as illustrated in the figure, indicates that the background droplet loading was relatively low, as evidenced by the data from the first 5 minutes. Five minutes into the measurement period, a single cough was introduced. This resulted in a significant increase in droplets being propelled into the suction device due to the strong airflow from the cough, as evidenced by a substantial spike at $t$ = 5 minutes. At this juncture, the suction device removed over 6,000 droplets per second for each size bin.

After the initial spike, the observations indicate a diffusion timescale of several minutes for these small droplets. The peak concentration of diffused aerosol in the environment occurs approximately 5 minutes post-cough event.

Our previous findings [9] suggest that an exponential decay function aptly represents the decrease in droplet nuclei. The calculated half-lives of the droplets for the configuration considered in [9] were 11.0, 6.4, and 2.9 minutes for droplet size bins of 0.3, 0.5, and 1.0 μm, respectively. In the second experiment, the artificial cough was repeated every minute following the initial cough at the 5-minute mark within the measurement domain. The droplet removal rate from this setup is presented in Fig 4.

The continuous generation of aerosols resulted in the suction device's steady removal of droplets. Initially, the droplet removal rate swiftly escalated to over 6000 droplets per second for all size bins.

In the final series of experiments, we contemplated four distinct experimental configurations. In these scenarios, the particle counter was positioned at angles of 0, 45, 90, and 135 degrees relative to the manikin. In contrast, the manikin and the suction canister device positions were fixed, as illustrated in Fig 2. As the particle counter probe was no longer inserted into the suction canister, the results represent the droplet count measurements from the particle counter in the enclosed environment instead of the suction removal rates by the canister. This means that the results represent the ambient droplet concentrations in these experiments.

Fig 5 displays droplet count measurements for selected locations of one and four.

The normalized aggregated droplet counts and volumes for all locations and size bins are depicted in Fig 6.

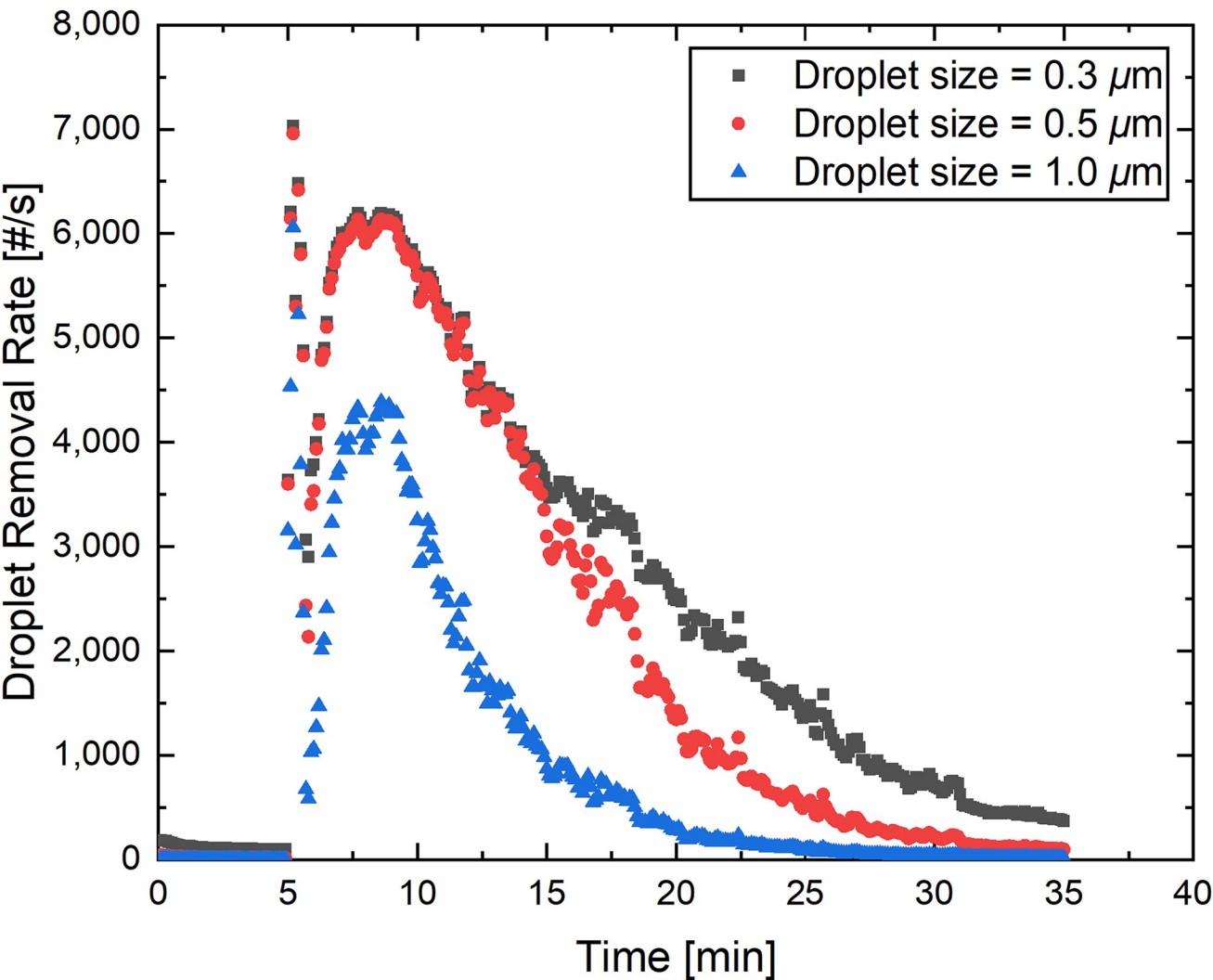

**Fig 3. Particle removal efficiency by the suction device as a function of time for size bins of 0.3, 0.5, and 1.0 from a sampling location inside the canister cavity.** A single cough was artificially generated and directed towards the suction canister from a distance of 50 cm. This cough event took place 5 minutes into the measurement period.

The normalized aggregate total droplet volume, calculated from the droplet counts and particle size bins, showed a similar trend. The suction device lessened the peak total droplet volume and accelerated its decay rate. These measurements indicate that the half-lives of the total droplet volume decreased from 23.6 minutes to 15.6 minutes with the application of the suction device. The aggregate peak droplet count was achieved approximately 8 minutes after the cough event. The peak droplet count with the suction device operational was about 10% lower. At 20 minutes, i.e., approximately 12 minutes after the peak droplet count, 82% of the peak droplet count remained suspended with the suction device off and 66% with the suction device on. However, since the peak droplet concentration was also reduced with the suction device, these percentages translate to an 18% removal rate without the suction device and a 23% removal rate with the suction device from their respective peak values. At 30 minutes or about 22 minutes after the peak droplet count, the reduction from peak counts was 24% without the suction device and 43% with the suction device.

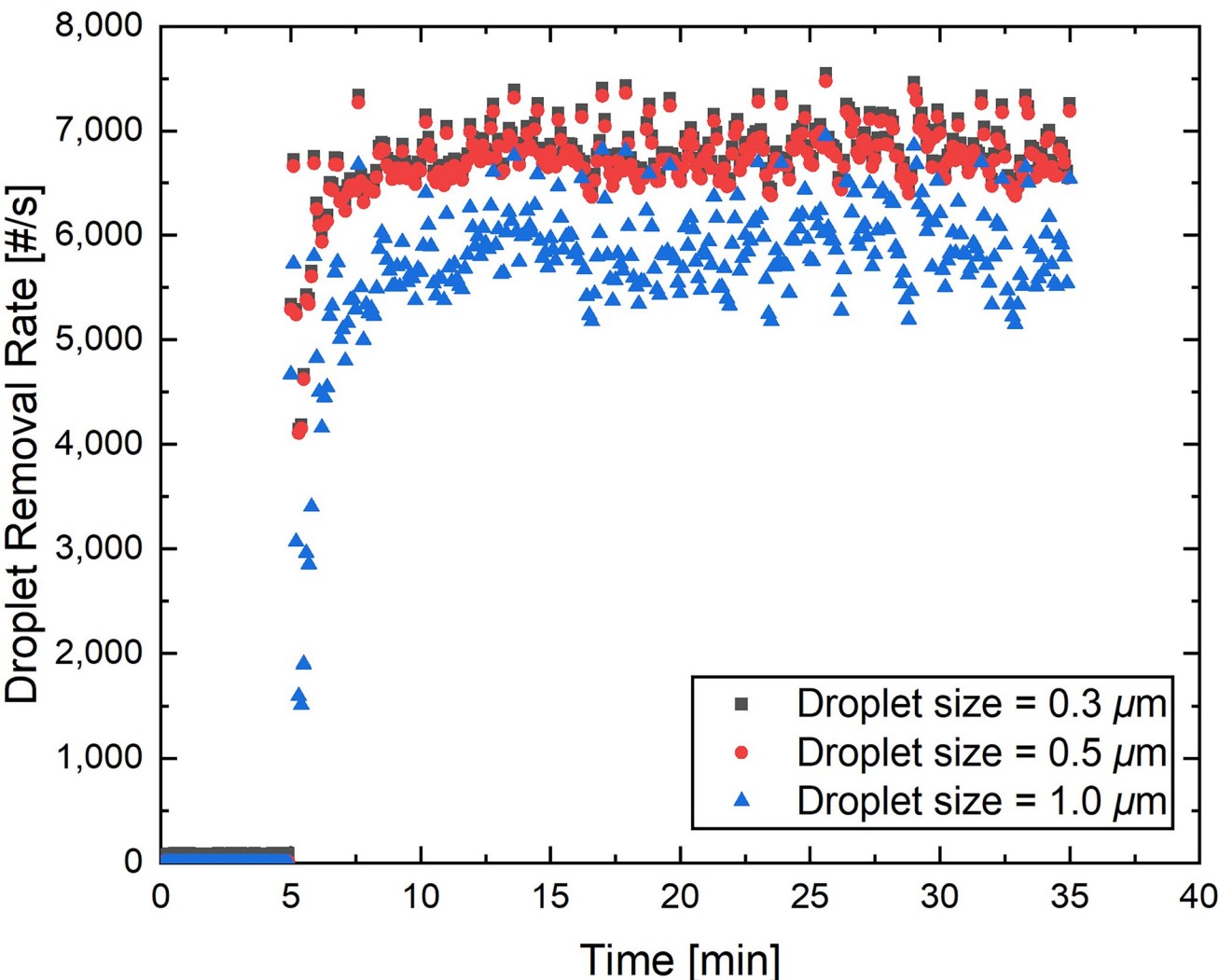

**Fig 4. Particle removal efficiency by the suction device as a function of time for size bins of 0.3, 0.5, and 1.0 from a sampling location inside the canister cavity.** An artificial cough was generated and directed towards the suction canister from a distance of 50 cm every minute. The initial cough event occurred 5 minutes into the measurement period.

## Discussion

The experiment's findings confirm the suction device's capability to effectively remove droplets from the environment, making it a vital tool in enhancing indoor air quality. Given the sustained performance of the suction device irrespective of single or multiple cough events, this demonstrates its potential utility in reducing the risk of airborne disease transmission.

This information further reinforces the idea that small droplets can remain airborne for extended periods in constrained spaces with limited air circulation. The calculated half-lives of the droplets of different size bins provide evidence of smaller droplets' increased air suspension capacity compared to the larger ones. This is consistent with our previous research [9] and related literature [10–12].

The results from the sustained cough in the second experiment (see Fig 4) highlight the robustness of the suction device, especially with its consistent performance over time. Figs 3

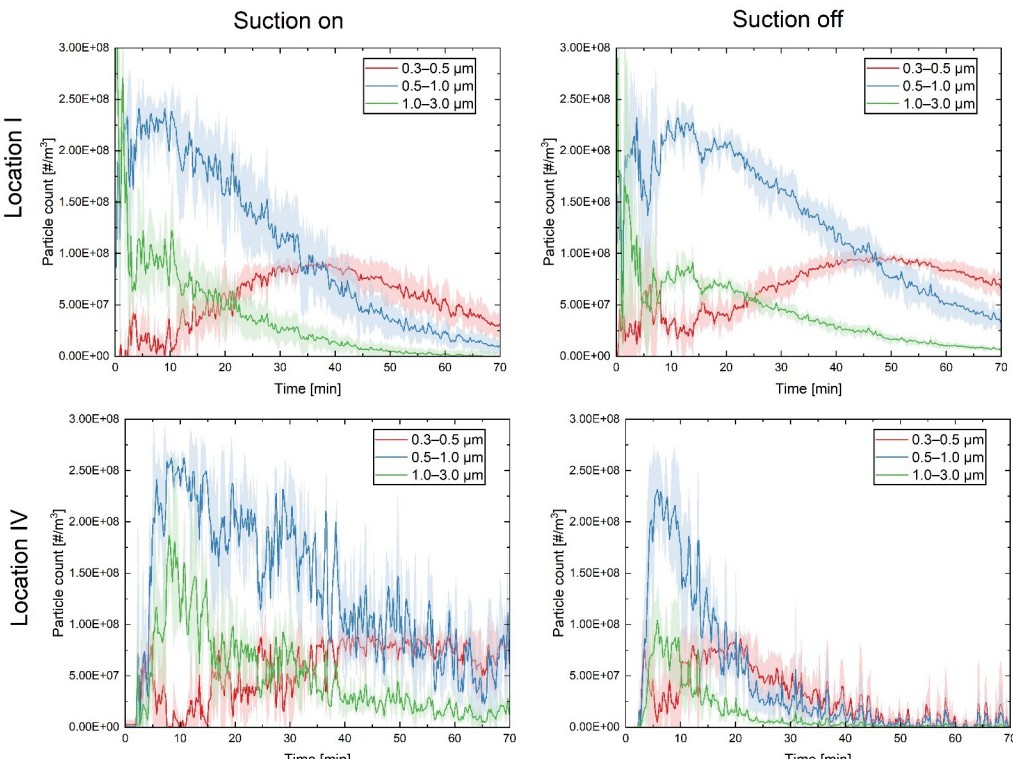

**Fig 5. Droplet nuclei count measurements for three size bins of 0.3, 0.5 and 1.0 μm from sampling locations 1 and 4.**
The shaded region represents the standard deviation of the average over five measurements. The top and bottom rows correspond to the droplet count measurements for locations one and four, respectively. The left column corresponds to particle counter data obtained when the canister was operational, and the right column represents data from when it was not in use.

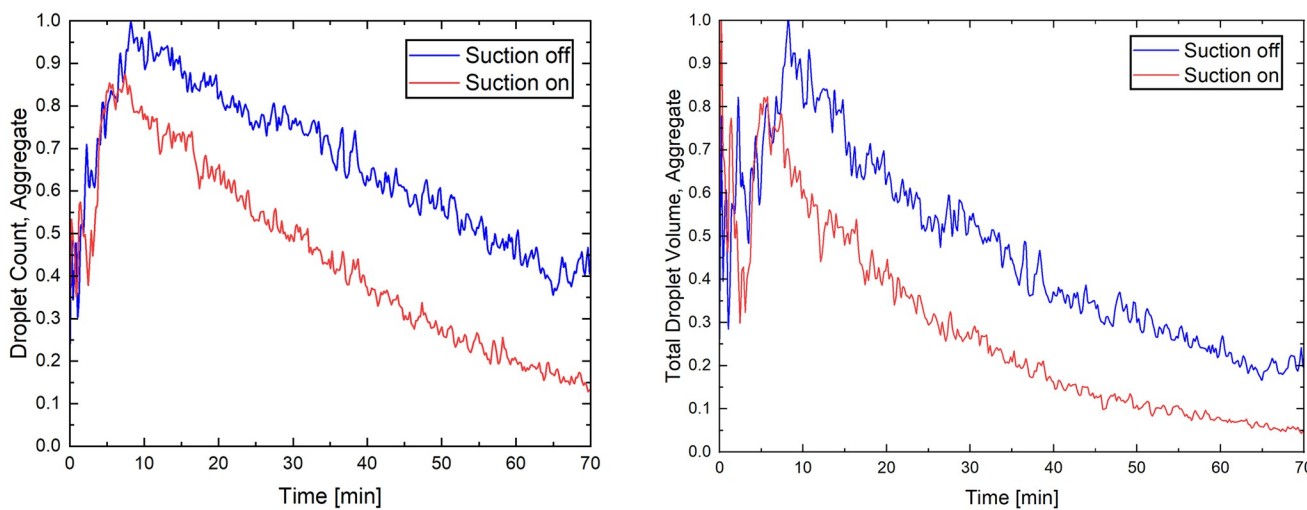

**Fig 6. Aggregate total droplet nuclei count measurements (left) and total droplet nuclei volume (right) for all size bins summed from all sampling locations.**

and 4 provide a comprehensive view of the suction device's performance under varying conditions. Regardless of whether it was handling a single cough event or coping with repeated events at regular intervals, the device demonstrated an ability to remove droplets from the environment efficiently. Given that the total viral load is likely proportional to the total aerosol volume [13, 14], the total viral loading will be reduced with the deployment of these devices. This makes it a promising tool for enhancing indoor air quality and potentially reducing the risk of airborne disease transmission. However, further study is required to confirm this.

For the series of experiments considering different experimental configurations, it is evident that the position and angle of the particle counter play a crucial role in the outcome. Notably, there is a significant difference in droplet counts when the particle counter was placed at different locations. Studies have indicated that cough droplets spread extensively sideways [15], with the turbulent flow of the initial puff cloud facilitating mixing. Despite some variability and noise in the measurements, a clear decay pattern was observed, irrespective of whether the canister was switched on or off. The canister reduced aerosol counts more quickly than background dissipation. Yet, at location 1, aerosol droplets persisted for up to an hour, albeit at reduced concentrations.

We also observed that the suction device proved more effective at location four. This could be attributed to location four being off-axis and positioned behind the manikin, implying that droplets could only reach this location via diffusion. Given that the cough was directed towards the suction device, it effectively eliminated particles diffusing in the opposite direction once the cough stream had halted.

The drop in the half-lives of total droplet volume with the device in operation underscores its efficiency. Analysis of the normalized aggregated droplet count reveals that the peak droplet counts typically occurred around the 7-minute mark on average. When the suction device was in operation, this peak count was found to be roughly 15% lower and appeared earlier than when no suction was applied. Even though droplets continued to exist, the rate at which droplet counts diminished increased noticeably compared to scenarios where no suction was utilized.

Our results are consistent with those of other similar devices. For example, K. Okuhata *et al.* [4] evaluated a tabletop HEPA filter device's effectiveness using particle image velocimetry. This study measured the planar velocity field and visualized aerosol particle dispersion. Results indicated that when used, the suction device reduced aerosol droplets by 91.8% and deposited droplets by 68.7%. However, caution should be exercised when comparing such measurements across studies, as the removal rates depend on several factors, including ambient conditions, cough composition, room size and geometry, and the measurement technique.

Several other studies have looked at portable local exhaust systems and extra-oral devices, primarily for dental, ENT and GI procedures [16–23]. Most studies show that portable devices can significantly reduce droplet counts. T. Maurais *et al.* [23] even demonstrated that portable devices perform better in isolation (compared to a combination of room ventilation and devices). The increase in turbulence from the negative pressure ventilation is thought to more easily disperse droplets that can be removed by a portable device close to the droplet source. Also, although studies confirm other designs, such as patient hoods or extra-oral suction devices, are effective, they are limited to certain clinical environments [16, 19–21, 24].

Other advantages of our design include:

- More portable [4, 25–30] and can be connected to most standard hospital suction ports

- Adaptable design that can be easily modified

- 3D printing makes it possible to scale up production quickly in the case of another respiratory pandemic

- It does not require the use of a HEPA filter (but can be easily adapted to include one)

- The design already provides space for UVC filters that can be added for decontamination

Finally, our results align with other studies demonstrating that portable suction devices can significantly reduce aerosol droplet concentrations, even in real-world environments [31–34], including homes, offices, schools and hospital rooms. Further study is warranted to evaluate the effectiveness of our device in other settings outside of the laboratory.

### Limitations and sources of error

Our study's primary sources of error stem from the optical properties of the aerosols generated by the simulated cough, the chaotic nature of cough flows due to turbulence, and the measurement accuracy of the particle counter. The particle counter used in this study has a sampling flow rate accuracy of ±5% and a size resolution of 15%. Its calibration is traceable to the National Institute of Standards and Technology (NIST). This counter employs a light scattering technique to determine particle concentration. This method introduces particles into an optical chamber, and their scattering intensity is recorded. The properties of the particles govern the correlation between scattered light intensity and particle concentration. In situations where these properties are not defined, as in this study, there might be significant deviations in the absolute particle counts. However, since our primary objective hinges on comparing particle counts against a baseline, the observed trends should remain consistent.

Furthermore, the nature of turbulence is chaotic, with its behaviours being predictable predominantly in a statistical sense due to its extreme sensitivity to starting conditions. This high degree of variability presents challenges in studying phenomena such as cough flows. To account for variations attributed to turbulence, we repeated experiments several times. Our findings are represented through mean and standard deviation values, as illustrated in Fig 5.

Further details on the robustness of the cough generator manikin, the measurement technique, and associated errors are discussed in [9]. In this reference, cough images captured from the manikin and spatial measurements taken without a suction device were compared with computational fluid mechanics analysis, incorporating humidity and temperature effects. The standard deviation values, shaded around the mean in Fig 5, highlight significant decay and trends that cannot be solely attributed to variations. However, it should be noted that inherent variability is an inevitable aspect of studies involving the chaotic nature of turbulence, which predominates cough flow dynamics. Consequently, measurements were repeated, and mean values were presented to ensure reliability and consistency in our findings.

### Conclusions

This study describes the design and fabrication of a suction canister from thermoplastic polyester using 3D FDM printing. The design and deployment of the device prioritized flexibility, speed, and affordability and could be used for rapid deployment in future pandemics.

We confirmed its ability to effectively remove droplets from the momentum-driven initial cough stream and the later droplet-filled ambient air. The device's suction efficiency also consistently maintained a stable droplet removal rate across multiple cough events, showcasing its reliable performance.

While the results did fluctuate depending on location, ambient conditions, and characteristics of the turbulent cough, the aggregated data demonstrated a substantial reduction in aerosol half-life, dropping from 23.6 minutes to 15.6 minutes.

This research highlights the potential of a straightforward, cost-effective device in mitigating airborne disease transmission. The scalability of this solution is impressive, as multiple

devices could be deployed rapidly and at low cost in public spaces, thereby improving public health safety. Its affordability and swift deployment via FDM 3-D printing make this device particularly well-suited for use in settings with limited resources.

## Acknowledgments

The authors also acknowledge the contributions from Karl Magtibay, BME PhD Student, Toronto Metropolitan University in creating 3D model drawings for the project.

## Author Contributions

**Conceptualization:** Ahmet E. Karataş, Steve Lin, Karthi Umapathy, Rohit Mohindra.

**Data curation:** Kai Lordly, Ahmet E. Karataş.

**Formal analysis:** Kai Lordly, Ahmet E. Karataş, Karthi Umapathy.

**Investigation:** Kai Lordly, Ahmet E. Karataş.

**Methodology:** Ahmet E. Karataş, Steve Lin, Karthi Umapathy, Rohit Mohindra.

**Project administration:** Ahmet E. Karataş.

**Software:** Ahmet E. Karataş.

**Supervision:** Ahmet E. Karataş.

**Visualization:** Ahmet E. Karataş.

**Writing – original draft:** Ahmet E. Karataş.

**Writing – review & editing:** Steve Lin, Karthi Umapathy, Rohit Mohindra.

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
