## [Decision Letter · Decision Letter 0]

19 Jan 2024

PONE-D-23-38824Effectiveness of a suction device for containment of pathogenic aerosols and dropletsPLOS ONE

Dear Dr. Mohindra,

Thank you for submitting your manuscript to PLOS ONE. After careful consideration, we feel that it has merit but does not fully meet PLOS ONE’s publication criteria as it currently stands. Therefore, we invite you to submit a revised version of the manuscript that addresses the points raised during the review process.

We look forward to receiving your revised manuscript.

Kind regards,

Sara Hemati

Academic Editor

PLOS ONE

Reviewers' comments:

Reviewer's Responses to Questions

**Comments to the Author**

1. Is the manuscript technically sound, and do the data support the conclusions?

Reviewer #1: Partly

Reviewer #2: No

2. Has the statistical analysis been performed appropriately and rigorously? 

Reviewer #1: No

Reviewer #2: No

3. Have the authors made all data underlying the findings in their manuscript fully available?

Reviewer #1: Yes

Reviewer #2: No

4. Is the manuscript presented in an intelligible fashion and written in standard English?

Reviewer #1: Yes

Reviewer #2: Yes

5. Review Comments to the Author

Reviewer #1: ‎1.‎ Abstract should be revised basically.‎

‎2.‎ Introduction is very general and need to be elaborative to explore the actual philosophy. ‎Authors have done through literature survey and have presented the past works. But, what ‎kind of innovation will be brought to the literature with this article? Therefore, the state-of-‎art should be clearly specified in detail in the Introduction part. Hypothesis should be given. ‎How this work is different from the available literature?‎

‎3.‎ The introduction of the manuscript should be analyzed the critical gap in the literature ‎and how the present study mitigates the gap.‎

‎4.‎ Please briefly clarify the problem statement, objectives, and future prospects in the ‎introduction section. Provide one nice and technically sound paragraph at the end of ‎introduction section about what is covered in the manuscript.‎

‎5.‎ The novelty carried out with this work was not reported and would be emphasized?‎

‎6.‎ Discussion” section should be revised basically. It is not suitable for a manuscript. ‎The ‎manuscript needs more adequate discussion with supporting latest references. ‎

‎7.‎ In the discussion section, the potential limitation of the study should be highlighted and ‎in ‎the conclusion, novel insight should be clearly highlighted.‎ ‎

‎8.‎ It is also recommended to discuss and explain what should be the appropriate policies based ‎on the findings of this literature.‎

‎9.‎ Please make sure your conclusions and future perspectives section underscores the ‎scientific value-added of your paper and/or the applicability of your results. ‎

Reviewer #2: There are many research papers already published on this topic and this ‎one ‎follows the same structure as many of the ‎previously ‎published articles, without adding any significance to the current knowledge. In my point of view, the work reported in this manuscript is not suitable ‎for ‎publication in this journal, since no conceptual or technological innovation is studied ‎and ‎presented. Thus, I reject this paper as the goals and importance of the journal do not allow ‎to ‎recommend this type of manuscript.‎

6. PLOS authors have the option to publish the peer review history of their article (what does this mean?). If published, this will include your full peer review and any attached files.

Reviewer #1: No

Reviewer #2: No

---

## [Author Response · Author response to Decision Letter 0]

4 Jun 2024

General Response to Statistical Analysis

In response to the reviewer's concerns regarding statistical analysis, our methodology and experimental design are an extension of the work presented in reference [9]. This previous research involved capturing cough images from the manikin and taking spatial measurements without a suction device. These were then analyzed alongside computational fluid dynamics simulations that considered the effects of humidity and temperature. Figure 5 presents the current manuscript's standard deviation and mean values, underscoring significant decay and trends that transcend mere variability. To further address the reviewer's concerns, we have added a new paragraph at the end of the "Sources of Error" section. 

Reviewer #1

The abstract has been revised. 

Response to Comments 2‒5: We recognize that our initial submission may not have sufficiently emphasized the literature gaps and the innovative contributions of our study. In response to the reviewer's perceptive observations, we have rewritten our Introduction and Discussion, detailing the gaps in the current literature that our study aims to bridge. These additions clarify the two primary gaps in the existing literature that our study seeks to fill. Firstly, we outline the need for a robust experimental method for the direct spatial and temporal assessment of droplet counts instead of relying solely on inferences from flow measurements obtained via particle image velocimetry and numerical simulations. Secondly, we highlight the absence of a cost-effective, rapidly deployable suction device adaptable to many indoor environments in the current literature. Finally, our results demonstrate the device's efficacy with multiple cough events, which is unique to our study design. We believe these revisions substantially strengthen the manuscript by providing a clearer exposition of its novel contributions and how it diverges from and builds upon the current state-of-the-art. 

The discussion has been revised to include more in-depth comments and supporting references, as well as more discussion about the limitations and the novel aspects of the research. 

The manuscript has been updated to include policy implications for this device.

The conclusions and future perspective section has been updated to highlight the significance of our work. 

Reviewer 2 

We believe this revised version of the manuscript highlights our work's unique value and innovation.

---

## [Editor Report · Decision Letter 1]

6 Jun 2024

Effectiveness of a suction device for containment of pathogenic aerosols and droplets

PONE-D-23-38824R1

Dear Dr. Mohindra,

We’re pleased to inform you that your manuscript has been judged scientifically suitable for publication and will be formally accepted for publication once it meets all outstanding technical requirements.

Kind regards,

Sara Hemati

Academic Editor

PLOS ONE
---

## [Editor Report · Acceptance letter]

2 Jul 2024

PONE-D-23-38824R1 

PLOS ONE

Dear Dr. Mohindra, 

I'm pleased to inform you that your manuscript has been deemed suitable for publication in PLOS ONE. Congratulations! Your manuscript is now being handed over to our production team.

Kind regards, 

on behalf of

Dr. Sara Hemati 

Academic Editor

PLOS ONE